# Enhancing Focusing and Defocusing Capabilities with a Dynamically Reconfigurable Metalens Utilizing Sb_2_Se_3_ Phase-Change Material

**DOI:** 10.3390/nano13142106

**Published:** 2023-07-19

**Authors:** Chen Shen, Jiachi Ye, Nicola Peserico, Yaliang Gui, Chaobo Dong, Haoyan Kang, Behrouz Movahhed Nouri, Hao Wang, Elham Heidari, Volker J. Sorger, Hamed Dalir

**Affiliations:** 1Department of Electrical and Computer Engineering, George Washington University, Washington, DC 20052, USA; imchucklate@gwu.edu (C.S.); ygui82@gwu.edu (Y.G.); chaobo17@email.gwu.edu (C.D.); movahhed@gwu.edu (B.M.N.); volker.sorger@ufl.edu (V.J.S.); 2Department of Electrical and Computer Engineering, University of Florida, Gainesville, FL 32611, USA; alcatraz@gwu.edu (J.Y.); npeserico@gwu.edu (N.P.); haoyan_kang@gwu.edu (H.K.); hwang40@gwu.edu (H.W.); el.heidari@ufl.edu (E.H.); 3Florida Semiconductor Institute, University of Florida, Gainesville, FL 32603, USA

**Keywords:** phase-change material, metalens, Sb_2_Se_3_, ITO, reconfigurable optics, microfabrication

## Abstract

Metalenses are emerging as an alternative to digital micromirror devices (DMDs), with the advantages of compactness and flexibility. The exploration of metalenses has ignited enthusiasm among optical engineers, positioning them as the forthcoming frontier in technology. In this paper, we advocate for the implementation of the phase-change material, Sb${}_{2}$Se${}_{3}$, capable of providing swift, reversible, non-volatile focusing and defocusing within the 1550 nm telecom spectrum. The lens, equipped with a robust ITO microheater, offers unparalleled functionality and constitutes a significant step toward dynamic metalenses that can be integrated with beamforming applications. After a meticulously conducted microfabrication process, we showcase a device capable of rapid tuning (0.1 MHz level) for metalens focusing and defocusing at C band communication, achieved by alternating the PCM state between the amorphous and crystalline states. The findings from the experiment show that the device has a high contrast ratio for switching of 28.7 dB.

## 1. Introduction

Since the 1990s, Digital Micromirror Devices (DMDs) have been commercially utilized in many fields such as projection systems, spectroscopy, and medical imaging, marking their significance as a form of microelectromechanical systems (MEMS) technology [1]. As the technology continues to evolve, DMDs have found increased applications in advanced areas, including 3D printing, 3D scanning, and emerging technologies such as augmented reality (AR) and virtual reality (VR) [2,3]. The core of DMD technology is an array of up to millions of tiny, individual, tiltable mirrors. Each of these micromirrors corresponds to one pixel in the projected or displayed image. These mirrors can be tilted in two directions, representing either an ‘on’ or an ‘off’ state. The rapid flipping of these mirrors, in conjunction with a light source and color wheel, generates the grayscale and color images that we see in digital light processing (DLP) displays or projectors [4]. Nonetheless, advancements in technology have highlighted several shortcomings of DMDs in both research and commercial applications, including issues in terms of size and weight, flexibility, and efficiency. Compared to other proposed on-chip integrated photonic hardware accelerators and in-memory computational photonic tensor cores, all-optical convolution solutions offer an alternative approach with a higher data processing rate and lower power consumption [5,6,7,8,9]. A 4f-type optical system has been utilized to introduce a method for optical matrix multiplication. This method leverages a computer-generated holographic mask placed at the co-focal plane between two lenses to perform matrix inner-product multiplication. The speed of light is leveraged to complete this multiplication in real time, thereby surpassing any duration necessitated by an electronic computer. Following two decades of advancement, the kernels produced by cutting-edge DMDs have superseded the earlier hologram mask, boasting a significantly enhanced throughput. This improvement is sufficient to accurately simulate entire neural planes, thereby making the prospect of optical convolutional efficiency increasingly feasible [10]. With the concurrent development of high-performance electro-optic modulators, photodetectors, signal converters, and reconfigurable metalenses, the realization of a fully integrated all-optical neuromorphic architecture is in sight [11,12,13,14,15].

The burgeoning field of metasurface technology has sparked significant innovation and advancements in the optical lens industry, particularly with the development of compact and miniaturized lenses known as metalenses [16,17,18,19]. These metalenses have emerged as a practical and efficient alternative to traditional bulk lenses due to their ability to control the wavefront of incident light [20]. By leveraging subwavelength antenna arrays designed using standard planar microfabrication technologies, metalenses offer precise control over the optical wavefront, enabling efficient manipulation of light at the nanoscale level [21,22]. One of the significant developments in metasurface technology is the shift in focus toward dynamic and reconfigurable meta-optics, leading to the creation of advanced reprogrammable varifocal lenses [23,24,25]. These lenses incorporate active metasurfaces, allowing for on-demand control and modification of their optical responses. Various mechanisms have been explored to achieve this active tuning capability, including mechanical deformation, electrochemical reactions, electro-optic and thermos-optic effects, and the use of phase-change materials (PCM) [26,27]. Notably, the use of chalcogenide alloy-based phase-change materials has gained attention for their nonvolatile active metasurface tuning capabilities, stemming from refractive index contrast achieved through crystal structural transitions [28,29,30,31,32,33]. The compact size and tunability of metalenses offer a wealth of potential applications in the field of nanophotonics. These include beam switching, lensing, dual-functional devices, and active holography, among others. Metalenses provide a flexible and compact solution for manipulating light at the nanoscale, opening up new possibilities in various fields, such as optical communications, imaging systems, and biomedical optoelectronics. Furthermore, the development of non-volatile reconfigurable metalenses represents a significant stride toward the realization of integrated all-optical free-space convolutional neuromorphic networks, which have the potential to surpass traditional devices like DMDs. In addition to their potential applications and advantages, the utilization of metasurfaces and metalenses holds great promise for imaging systems and microscopy [34]. The precise control of optical wavefronts achieved by metasurfaces enables the generation of arbitrary wavefronts and the implementation of traditional optical components in a compact and lightweight form factor. This advancement has the potential to revolutionize imaging systems by reducing their size and complexity while maintaining high-resolution imaging capabilities. By pushing the boundaries of miniaturization and enhancing the efficiency of light manipulation, metasurfaces offer exciting prospects in the fields of integrated optics and advanced reconfigurable optical systems [35].

Our work was influenced by Mengyun et al.’s research, where they effectively used the ultralow-loss PCM Sb${}_{2}$Se${}_{3}$ [36]. Their proposed concept includes a varifocal metalens that operates at near-IR wavelengths, which they successfully demonstrated through experimentation. They created a unique all-dielectric layer composed of Sb${}_{2}$Se${}_{3}$-based nanopillars with a moderately low aspect ratio that is adequate to sustain Mie-type resonances of both electric and magnetic modes. This resulted in full 2π optical phase control while also minimizing transmission loss. They showcased a single focal metalens that displayed high focusing efficiency and superb intensity tunability. Furthermore, their varifocal metalens exhibited diffraction-limited performance and similar focusing efficiency at both focal planes. However, their paper did not include an on-chip PCM switching method. They did suggest in the discussion that integrating an on-chip heater could potentially lead to highly efficient and dynamically tunable metalens. In this study, we propose and demonstrate a reconfigurable lens based on the optical property transformation of the low-loss phase-change material Sb${}_{2}$Se${}_{3}$. This transformation, denoted by n, occurs between amorphous and crystalline states and is induced by a novel, low-loss transparent heater based on ITO. This work serves as a proof-of-concept for this promising approach.

## 2. Result

### 2.1. Design

The phase profile φ(R) of the metasurface needs to be precisely digitized and structured as a hyperbolic form. This precise configuration guarantees a focus spot limited only by diffraction:(1)$$\varphi \left(R,\lambda \right)=-\frac{2\pi }{\lambda }\left(\sqrt{R^{2}+f^{2}}-f\right)$$*R* refers to the radial position on the metasurface, and *f* symbolizes the focal length. Equation (Equation 1) illustrates the necessary condition that light, which is incident normally and has a wavelength λ, should accumulate at the focus coherently.

Phase-change materials can be transitioned between crystalline and amorphous states when subjected to specific thermal conditions [37,38]. Here we select Antimony Triserenide (Sb${}_{2}$Se${}_{3}$) as the modulation unit due to its minimal absorption loss, coupled with a high refractive index contrast in the 1550 nm telecommunication spectral range [39]. The unique characteristic of this hybrid metasurface lies in its non-destructive capacity (compared to the conventional optical lens) to alternate the focal length of a silicon photonic metalens between two distinct values, a capability made possible by the phase-change material Sb${}_{2}$Se${}_{3}$. This is achievable as the refractive index of silicon sits between the two alterable states of the optical phase-change material. The fact that Sb${}_{2}$Se${}_{3}$ maintains its transparency in both states facilitates the creation of nearly phase-only metasurface structures.

A variety of methodologies for controlling PCMs have been proposed, including direct laser Joule heating, graphene micro heaters, doped silicon heaters, and low-loss metal micro heaters placed on both sides. Our heterostructure design leverages the incorporation of a robust, low-absorption, resistive, transparent microheater, Indium Tin Oxide (ITO) [40,41,42,43,44,45]. It is achievable through meticulous production control and post-processing procedures such as annealing. ITO is well-known for its durability and transparency, making it an excellent choice for creating microheaters. In our design, the ITO microheater is separated from the metalens, which brings about several advantages. This separation ensures that the microheater does not interfere with the optical performance of the metalens, thereby preserving high transmission efficiency. Additionally, this design strategy helps avoid potential damage to the metalens that might occur from thermal effects, thus enhancing the overall robustness and longevity of the system. In this way, our design serves as a solid model for achieving high transmission efficiency in optical systems [46]. The design procedure of the reconfigurable metalens with a dimension of 1.3×1.3 mm${}^{2}$ is shown in Figure 1a. We deposit a thin layer of ITO on a glass substrate to serve as a micro-heater, heating the PCM pillar to enable lens tuning. The proposed metalens pillar is then constructed by patterning a Sb${}_{2}$Se${}_{3}$ film atop the ITO. The focusing ability of the metalens is determined by phase modulation profiles, arranged in a hyperbolic phase profile array as depicted in Figure 1d, to attain the desired initial focusing distance. The refractive index of Sb${}_{2}$Se${}_{3}$, with its substantial difference (Δn) and exceptionally low-absorption coefficient κ, facilitates the modulation of the meta pillar’s refractive index. This modulation allows us to alter the phase profile, thereby defocusing the lens or switching to a different focal distance while maintaining high transmission efficiency in the amorphous state. The optical image of the metalens is shown in Figure 1b and the SEM image is shown in Figure 1c.

To align with the computed phase map for focusing, we study the phase-change effect by varying the radius and period parameters of the cylindrical nanopillars in both the amorphous and crystalline states, as demonstrated in Figure 2a. Based on the phase map design, we select nanopillars of appropriate dimensions and arrange them on the ITO heater, facilitating the conversion of transmitted light at the predetermined focal distance.

The first step in our process involves the sputtering of a thin, low-loss ITO film onto a glass substrate, creating a heating element. Following this, we sputter and pattern Sb${}_{2}$Se${}_{3}$ nanopillars using a chlorine-based etch. Subsequently, a thin layer of silicon oxide is deposited onto the stack to form a protective capping layer. The tuning of Sb${}_{2}$Se${}_{3}$ for a phase transition relies heavily on the use of ITO, a low-absorption transparent conductive material, which is crucial for generating sufficient heat without contributing additional transmission loss. Through careful post-processing of the initially deposited ITO thin film, we were able to customize the ITO to achieve remarkably low loss and high resistivity. We performed ellipsometry on the ITO, and as shown in Figure 2b, the refractive index (n) exhibits an increase after a 30-min rapid thermal annealing (RTA). Simultaneously, there is a notable decrease in the extinction coefficient (k).

### 2.2. Simulation

A simulation is necessary to be conducted to test the feasibility of the concept, and we could further employ the customized settings as the preliminary configuration for the experiment. COMSOL Multiphysics 6.1 is a powerful, flexible software environment that allows users to model engineering-based physics phenomena. One of the many tools it offers is an electro-thermal simulation, which is used to investigate the interactions between electrical and thermal phenomena, like the ITO heating case in this project [47]. Electro-thermal simulations in COMSOL are based on coupling the electric currents and heat transfer in solids interfaces. This modeling process often involves applying electric currents or voltages to the structures in the model, which then generates heat due to Joule heating (also known as resistive or ohmic heating). This, in turn, can affect the electrical properties of the material due to the fact that many materials’ electrical properties are temperature dependent. To perform an electro-thermal simulation, we defined both the electrical and thermal properties of the materials involved. Then, we set up the electrical conditions, like current or voltage, apply thermal conditions, such as convection or radiation, and then solve the coupled physics problem. COMSOL allows users to analyze the results in 1D, 2D, or 3D, providing a comprehensive understanding of how electric and thermal properties interact in your specific application. In this project, considering the thickness of layers is not negligible, we selected 3D simulation.

The structural geometry is defined using the “work plane” and “extrude” functions. We start by creating the glass substrate, upon which a 200 nm-thick ITO layer is overlaid. The ITO parameter is referred from a published paper [48]. The electrical conductivity is $1.3\times 10^{6}$ S/m, the heat capacity at constant pressure is 341 J/(kg·K), the relative permittivity is 3.34 and the density is 7120 kg/m while the thermal conductivity is 0.21 W/(m·k). Following this, two 50 nm-thick Ti/Au alloy contact pads, employed for grounding and applying electrical potential, are positioned atop the ITO layer. The initial temperature of both the structure and the environment is preset at 293.15 K. One pad’s electrical potential is then adjusted to 5 V. We proceed to set the computation output time to 120 µs, with a time step of 1 µs. The simulation we conducted confirmed that our proposed heating platform could reach the necessary temperature range (500–900 K) to facilitate the phase transition of the PCM. Also, in COMSOL’s result section, we create a “cut point 3D” and set its xyz axis value to represent a center point of the ITO layer. Then we use the “point graph” function in the “1D plot group” to plot the temperature versus time figure. As shown in Figure 3c,d, the pulse duration used to heat the ITO heater layer for the processes of crystallization and amorphization are 2 µs and 8 µs, respectively. As for the cooling duration, the cooling down times of the device from 500 K and 900 K to below 350 K are estimated to be 50 µs and 90 µs, respectively.

### 2.3. Experiment

#### 2.3.1. Device Fabrication

A 200 nm-thick ITO layer is deposited on a glass base through a DC sputter process, which entails the evaporation of constituent materials [49]. To mitigate contact resistance upon application of external voltage, two contact pads composed of a Ti/Au alloy are strategically positioned at the periphery of the ITO layer. To ensure a homogenous coating, a 300 nm-thick layer of Sb${}_{2}$Se${}_{3}$ is meticulously sputter-deposited onto the ITO/SiO${}_{2}$ substrate, facilitating even distribution of the Sb${}_{2}$Se${}_{3}$ material across the substrate’s surface. In the following step, a layer of the negative photoresist (ARN 7520.18) is applied to the ITO-coated substrate, serving as a protective layer that is selectively exposed in the ensuing processes. The fabrication of the envisaged design is achieved utilizing the EBL Voyager system, which boasts high-resolution capabilities, accommodating write fields of up to 500 microns with a stitch error of less than 30 nm. Seamless lithography can be realized through fixed-beam-moving-stage (FBMS) technology. Post-exposure, the sample is subjected to a development process using the MF319 solution. This step removes the unexposed photoresist, thereby revealing the desired pattern on the substrate. To enhance structural refinement, the unprotected portions of the Sb${}_{2}$Se${}_{3}$ material—those not shielded by the patterned photoresist—undergo etching using Deep Reactive Ion Etching (DRIE) systems [50]. The DRIE process utilizes reactive gases and ion bombardment, effectively eliminating the unprotected Sb${}_{2}$Se${}_{3}$ material and leading to the creation of an intricately detailed structure that conforms to the intended design. In the final step of this fabrication process, a thin and transparent SiO${}_{2}$ layer is deposited to encapsulate the device, offering both physical protection and functional enhancement.

Here we employ three equations to calculate the power loss of light propagated through the ITO layer and glass substrate:(2)α=4πκ/λ
(3)$$I=I_{o}\times exp\left(-\alpha d\right)$$
(4)$$A=-10\times log_{10}\left(I/I_{o}\right)$$
The Equation (Equation 3) is the Beer-Lambert Law. Where I is the incident intensity, *I*${}_{o}$ is the transmitted intensity, α is the absorption coefficient and κ is the extinction coefficient of the material. A represents the Attenuation (in dB) of the light travel through the given design. After incorporating the design parameters into the equations, we calculate the attenuation for both the ITO layer and the glass substrate to be −0.322 dB.

#### 2.3.2. Experimental Setup

Our experimental setup scheme is shown in Figure 4. A fiber-coupled Fabry-Perot laser (FPL1009P, Thorlabs) driven by a compact laser diode driver (CLD1015, Thorlabs) with a 1550 nm wavelength and a 100 mW output power is utilized to meet the needs of the rapidly growing field of long-distance free-space optical communication, where the initial laser beam is subsequently linked to an optic fiber collimator. This is succeeded by a free-space beam expander to ensure the distribution of the light beam across the full active area of the Sb${}_{2}$Se${}_{3}$ nanopillars. After propagating 300 mm, the incident expanded beam is projected onto the metalens, which is held vertically by a mounted stage to comply with the transparent transmission requirement. A 5 V voltage is applied to the Ti/Au contact pad attached to the ITO to achieve the evenly distributed 500 K temperature, where nanopillars switch from the initial amorphous state to the crystalline state, realizing the phase tuning to focus the incident beam. With another slightly longer pulse voltage, nanopillars, after reaching 900 K, switch from the crystalline state back to the amorphous state, tuning the phase to defocus the beam. When nanopillars are in their initial amorphous state, a 1550 nm high-speed camera is placed at the focal plane to record the intensity profile. After applying the pulse on the Ti/Au contact pad affixed to the ITO, the camera then captures and saves the intensity profile for the crystalline state. Correspondingly, the camera is placed at the focal plane of the crystalline state and records the intensity profile for both states. Due to the limitation on power measurement for a single focused beam out of a total of eight meta lens samples, multiple images for each state is recorded and its pixel value is averagely measured and normalized via Python script.

#### 2.3.3. Focusing and Defocusing

First, we measure the initial passive metalens with nanopillars in an amorphous state, where the camera is placed at its focal plane with an amorphous state’s focal length of 8 mm ($f_{1}$). The measured normalized intensity of the focusing beam in an amorphous state is shown as a red curve in Figure 5a, and the intensity profile is shown in Figure 5c. A rectangular pulse with 5 V and 2 µs duration (preliminary configuration from COMSOL simulation) generated by Keysight 33,520 B—Trueform Waveform/Function Generator is introduced to the Ti/Au contact pad affixed to the ITO, resulting in a uniform temperature distribution of 500 K temperature. This thermal condition prompts the nanopillars to transition from an amorphous to a crystalline state. The measured normalized intensity of the defocusing beam in the crystalline state at $f_{1}$ focal distance is shown as a blue curve in Figure 5a, and the camera-captured image is shown in Figure 5d. After the recording, the camera is replaced at the focal plane where the nanopillars are in the crystalline state with a focal length of 10 mm ($f_{2}$). The measured normalized intensity of the focusing beam in the crystalline state is shown as the blue curve in Figure 5b, and the intensity profile is shown in Figure 5f. Meanwhile, the measured normalized intensity of defocusing beam in an amorphous state at $f_{2}$ focal distance is shown as a red curve in Figure 5b, and the intensity profile is shown in Figure 5e. Subsequently, another slightly longer pulse (5 V, 8 µs) voltage pulse is applied to increase the nanopillars’ temperature to 900 K. This temperature change triggers a switch in the nanopillars from the crystalline state back to the amorphous state, altering the phase to disperse the incident beam.

The outcomes from the experiment are in agreement with the results from the COMSOL simulation, as we used the preliminary configuration from the COMSOL simulation and successfully captured images that demonstrate the anticipated focusing and defocusing.

The switching contrast ratio of the two states is determined as followed:(5)$$CR=10\log_{10}\left(\frac{P_{1,a}}{P_{2,a}}\times \frac{P_{1,c}}{P_{2,c}}\right)$$
where $P_{1,(2),a(c)}$ denotes the focused optical power at focal spot 1(2). We experimentally measured the focused power ratio between the true and phantom focal spots, yielding $P_{1,a}/P_{2,a}=16$ and $P_{1,c}/P_{2,c}=46$, which the corresponding CR equals to 28.7.

#### 2.3.4. Cyclability Test

We assessed the durability of our device through a series of “focusing-defocusing” cycles, without conducting an image readout until a predetermined cycle count was reached. Notably, there is no degradation observed after over 10,000 switching cycles, Hence, we assert that our device exhibits a cyclability of more than 10,000 cycles.

## 3. Discussion

One of the challenges associated with the metalens is the efficiency of heat dissipation on the ITO heating layer. The process of heating the nanopillars occurs rapidly, in several µs, which is advantageous for quick tuning. However, a significant drawback is the extended time required for the array of nanopillars to cool down (50–90 µs). When heat is applied to the nanopillars, they can quickly reach the desired temperature, enabling immediate changes in the metalens’ properties. However, once the heat source is removed, the densely packed nanopillars take a considerably longer time to return to their baseline temperature. This is due to the compact design of the metalens, in which the space between the layers is minimal. The extended cooldown period can potentially slow down processes or experiments that require multiple cycles of heating and cooling. Consequently, this aspect could limit the metalens’ performance in applications that demand rapid, repetitive changes. Improving the heat dissipation efficiency of the ITO heating layer is a critical area for future research and development to enhance the overall functionality of the metalens.

In conclusion, we employ a low-loss phase-change material, Sb${}_{2}$Se${}_{3}$, heated by low-loss conductive oxide material ITO layer in our efforts to create a reconfigurable varifocal metalens, capitalizing on the significant index contrasts and lossless attributes of these materials. In a proof-of-concept demonstration, we propose, fabricate, and assess an active metalens with the ability to focus and defocus by heating the nanopillars to different temperatures, thereby shifting their state from amorphous to crystalline and vice versa. The most noteworthy aspect of these findings is the impressive switching contrast ratio, which reaches as high as 28.7 dB. The study establishes that active metalenses, which are non-mechanical, can realize the focusing and defocusing functions like conventional high-precision bulk optics which use mechanical moving components. This opens up a new array of promising applications that completely capitalize on the advantages of Size, Weight, Power, and Cost (SWaP-C) associated with active metasurface optics. These applications could span imaging, sensing, display, and optical ranging.

## Figures and Tables

**Figure 1 nanomaterials-13-02106-f001:**
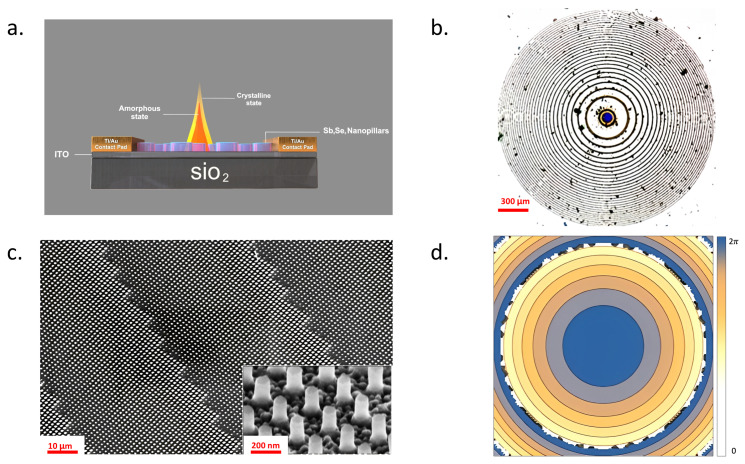
(**a**) Schematic of reconfigurable metalens with ultra-loss phase-change material Sb${}_{2}$Se${}_{3}$. (**b**) Optical image of the metalens. (**c**) Scanning Electron Microscopy (SEM) image of the metalens at varying levels of magnification. (**d**) 2D phase maps corresponding to amorphous state.

**Figure 2 nanomaterials-13-02106-f002:**
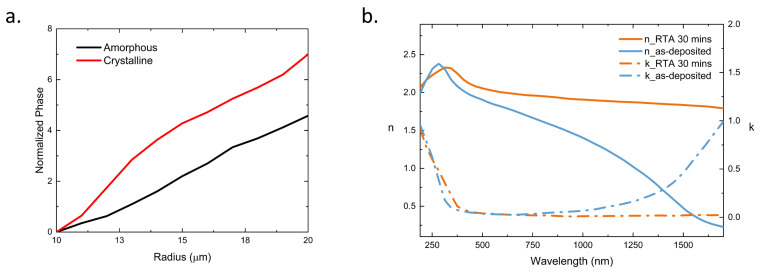
(**a**) Effective phase of meta pillar as radius changes at amorphous and crystalline state (**b**) Ellipsometry data of ITO before or after custom processing.

**Figure 3 nanomaterials-13-02106-f003:**
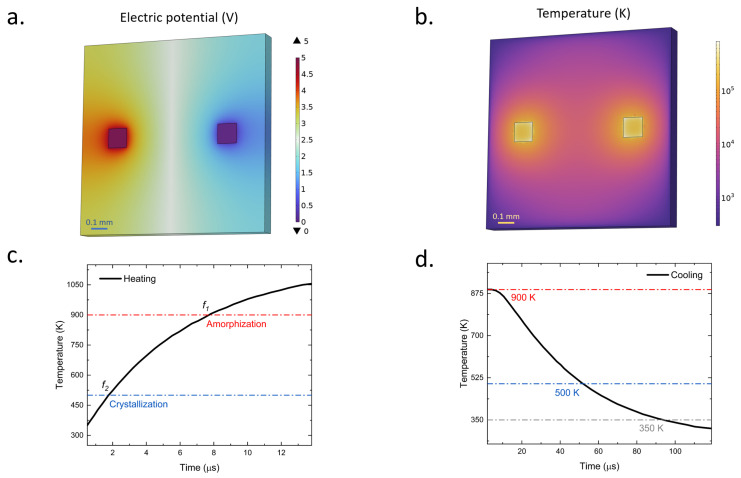
(**a**) Schematic representation of the electric potential scheme, demonstrating an applied voltage of 5V. (**b**) Heat distribution map on the ITO plane, highlighting uniform heat dispersal. (**c**) Time-lapse temperature plot during the heating phase. (**d**) Time-lapse temperature plot during the cooling phase.

**Figure 4 nanomaterials-13-02106-f004:**
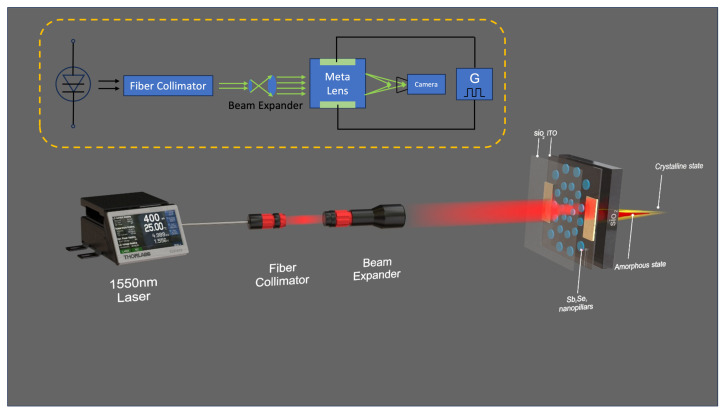
Illustration of the experimental setup used for phase modulation of Sb${}_{2}$Se${}_{3}$ nanopillars, showcasing the transition from an initial amorphous state (defocusing), through the crystalline state (focusing), and reverting back to the amorphous state (defocusing).

**Figure 5 nanomaterials-13-02106-f005:**
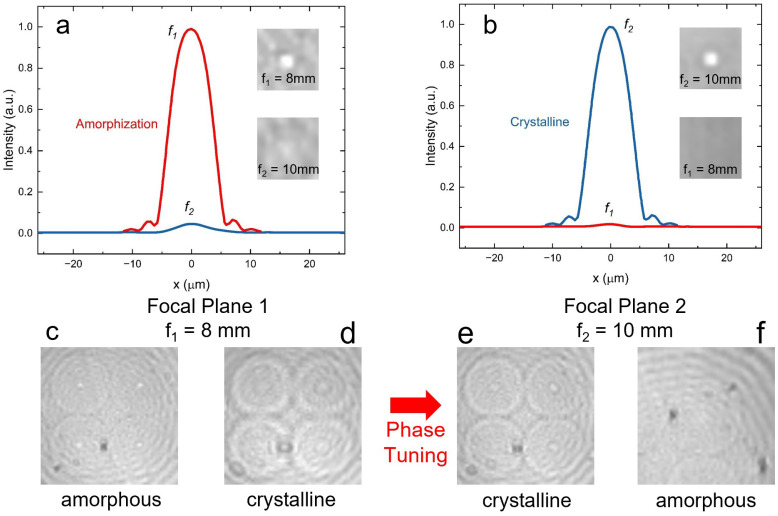
Tuning from amorphous to crystalline state. (**a**) The normalized power intensity of two states on focal plane 1. (**b**) The normalized power intensity of two states on focal plane 2. (**c**–**f**) The CCD camera captured images of metalens in different states at different focal planes.

## Data Availability

The main data supporting the findings of this study are available within the article. Extra data are available from the corresponding authors on reasonable request.

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
