# Peer review of "Enhancing Focusing and Defocusing Capabilities with a Dynamically Reconfigurable Metalens Utilizing Sb2Se3 Phase-Change Material"

_nanomaterials, 2023, doi:10.3390/nano13142106_

Round 1
Reviewer 1 Report
The authors present on a dynamically reconfigurable meatless with a tunable focal length. Experimental results are presented that demonstrate the performance of the device. The paper is well written and can be accepted for publication.
Author Response
Thank you for your endorsement and approval of this research and submission. Your recognition and support mean a great deal to us, and we deeply appreciate your positive response. We have put considerable effort into refining and extending this work to ensure its quality and relevance. Your validation further motivates us to continue our pursuit of excellence in our research endeavors. Once again, we express our gratitude for your appreciation and encouragement.
Reviewer 2 Report
This paper is about the switchable metalens using sb2se3 PCM.
I don't know what is the exact findings or contributions of this paper.
I could find a relevant papers just by googling "metalens Sb2S3".
Adv. Sci. 2023, 10, 2204899, DOI: 10.1002/advs.202204899
Varifocal Metalens Using Tunable and Ultralow-loss Dielectrics
Mengyun Wang, June Sang Lee, Samarth Aggarwal, Nikolaos Farmakidis, Yuhan He,Tangsheng Cheng, and Harish Bhaskaran*
It is needed that the author should clarify what is the main findings and contribution of the paper compare to above paper.
About the introduction, I couldn't fully understand the author's intention.
Especailly, through lines 75-92, what is the relation with the submitted paper?
I suggest that authors reorganize or rewriting the introduction section for the clear background of the paper.
Belows should be checked for the better understanding.
8: rapid? I don't see any number of speed of this reconfigurable metales. Shold mention about the exact switching speed.
78: our?
86: we?
105: missing 'Lanmbda' after wavelength
108: 'low' might be deleted
110-111: ??. If there are destructive for metalens, please refer and compare.
130: parenthesis, Fig.1(a)
155: For simulation, if there is any simulatiion result, authors should include it. Are simulation results used in experiments? Is it matched? Did the authors change the design using simulation for the necessary temp? More detail description will be helpful to understand why the authors bother to do some simulation.
efficiency? speed? any cross talk? side-effect?
236: contrast?
237: Should mention about the measured power for each case, not just ratio.
242: rapidly? Should mention the measured number of heating/cool down speed.
263: How come the authors can say that "attaining a optical quality that matches conventional high-precision bulk optics..." Need back data of this claim. Only high contrast ratio doesn't cover the claim.
Reviewer 3 Report
In this paper, authors have proposed a phase change material based reconfigurable flatlens and experimentally demonstrated its focusing and defocusing capabilities. The work is interesting, and the paper is written well. I have some concerns which currently prevent me from recommending publication in Nanomaterials.
1. Authors stated that the dimension of the metalens is 1.3 × 1.3mm2. I am wondering how uniform the temperature distribution in this area is for efficient reversible electrical switching. I suggest authors provide simulation or thermal imaging data to support this claim.
2. It seems that the crystallization and melting temperature of Sb2Se3 is above 200C and 611C, respectively. Is it possible to get this higher temperature using ITO heaters? Authors can provide microheater calibration data or some other data to support this claim.
3. Can authors comment on the switching cycle of the proposed device?
Round 2
Reviewer 2 Report
The authors have suitably revised the manuscript. I have no further comments here.